# Plant-Based Vaccines in Combat against Coronavirus Diseases

**DOI:** 10.3390/vaccines10020138

**Published:** 2022-01-18

**Authors:** Benita Ortega-Berlanga, Tomasz Pniewski

**Affiliations:** Institute of Plant Genetics, Polish Academy of Sciences, Strzeszyńska 34, 60-479 Poznań, Poland; tpni@igr.poznan.pl

**Keywords:** coronaviruses, COVID-19, MERS-CoV, SARS-CoV, biopharming, plant-based vaccines

## Abstract

Coronavirus (CoV) diseases, including Middle East Respiratory Syndrome (MERS) and Severe Acute Respiratory Syndrome (SARS) have gained in importance worldwide, especially with the current COVID-19 pandemic caused by SARS-CoV-2. Due to the huge global demand, various types of vaccines have been developed, such as more traditional attenuated or inactivated viruses, subunit and VLP-based vaccines, as well as novel DNA and RNA vaccines. Nonetheless, emerging new COVID-19 variants are necessitating continuous research on vaccines, including these produced in plants, either via stable expression in transgenic or transplastomic plants or transient expression using viral vectors or agroinfection. Plant systems provide low cost, high scalability, safety and capacity to produce multimeric or glycosylated proteins. To date, from among CoVs antigens, spike and capsid proteins have been produced in plants, mostly using transient expression systems, at the additional advantage of rapid production. Immunogenicity of plant-produced CoVs proteins was positively evaluated after injection of purified antigens. However, this review indicates that plant-produced CoVs proteins or their carrier-fused immunodominant epitopes can be potentially applied also as mucosal vaccines, either after purification to be administered to particular membranes (nasal, bronchus mucosa) associated with the respiratory system, or as oral vaccines obtained from partly processed plant tissue.

## 1. Introduction

Coronaviruses (CoVs) are clinically relevant pathogens that infect humans, livestock, mice, birds and many other wild animals [1]. They cause localized infections in the respiratory and/or intestinal tracts, in the liver and the central nervous system of their hosts [2]. They belong to the order *Nidovirales*, the family *Coronaviridae*. Based on phylogenetic analysis, CoVs are divided into four genera: the alpha, beta, gamma, and delta coronaviruses [2]. This family of viruses has gained in clinical relevance since 2003, when a new human coronavirus (SARS-CoV-1) was responsible for Severe Acute Respiratory Syndrome (SARS). Later in 2012, a new outbreak of another coronavirus (MERS-CoV) emerged in Saudi Arabia, causing Middle East Respiratory Syndrome (MERS) [3,4]. Most recently, in December 2019 in Wuhan, the Hubei Province, China, the pathogen responsible for a mysterious pneumonia was identified as SARS-CoV-2 and defined as the causal agent of Coronavirus Disease 2019 (COVID-19) [5]. Due to the rapid spread of the virus around the world, COVID-19 was declared to be a pandemic by the World Health Organization (WHO) in March 2020. The origin of this outbreak was associated with the Wuhan Wholesale Seafood Market, where not only seafood is traded, but also exotic fauna [6]. Due to the fact that the greatest diversity of CoVs has been found in bats [7], the hypothesis has been proposed that more recent CoV introductions to humans were originally bat viruses that propagate to an intermediate host (e.g., the Himalayan palm civet for SARS-CoV-1 and the dromedary camel for MERS-CoV), which then exposed humans to the viruses. According to the WHO, the SARS-CoV-1 pandemic affected 8096 people while the MERS pandemic affected 2494 people, with a fatality rate of 9.19% and 34.4%, respectively [8]. Until November 2021, emerging SARS-CoV-2 variants and successive waves of the COVID-19 pandemic have affected more than 261 million people worldwide and have caused more than 5.2 million deaths [9]. The exponential spread of SARS-CoV-2 together with the emerging outbreaks of new CoVs in recent years highlights the urgent need to develop effective therapies and vaccines against these pathogens.

## 2. Genome and Structure of Coronaviruses

CoVs are enveloped RNA viruses, ranging from 80 to 160 nm in diameter. Their genome is composed of non-segmented, positive-sense RNA, which contains from 27,000 to 32,000 nucleotides, and these are the largest genomes among mammalian viruses [10]. The genome of CoVs contains a 5′-cap structure and a 3′-poly-A tail; its organization is as the follows: 5′-leader-UTR-replicase-S (Spike)-E (Envelope)-M (Membrane)-N (Nucleocapsid)-3′ UTR-poly (A) tail [1]. The CoV genomes usually contain at least six open reading frames (ORFs). Two-thirds of the genome comprise two overlapping ORFs: ORF1a and ORF1b. Translation of these ORFs produces two large polypeptides: pp1a and pp1ab, which after undergoing co- and post-translational modifications result in 16 non-structural proteins (nsps) that play important roles in the viral replication process [11]. The rest of the genome codes four main structural proteins: spike (S), membrane (M), envelope (E), and nucleocapsid (N) proteins (Figure 1). These are essential for virion assembly and infection [12]. The S protein is a very large (~150 kDa) transmembrane and highly glycosylated protein that assembles into trimers on the viral envelope, and mediates attachment and fusion to the cell host [13]. The M protein is a small (~25–30 kDa) multispanning membrane protein, which contains a little N-terminal ectodomain and a cytoplasmic tail. It is the most abundant structural protein in the virion [14]. This protein promotes membrane curvature, gives the virion its shape and allows the connection to the nucleocapsid [1]. In turn, the E protein is a small (~8–12 kDa) hydrophobic transmembrane protein that is involved in virus assembly, release and viral pathogenesis [15]. The N protein (43–50 kDa) is a phosphoprotein which contains three regions: an N-terminal domain, a RNA-binding domain and a C-terminal domain. It is the main structural component of the nucleocapsid and can bind to RNA to protect it from degradation by ribonucleases [16]. N is also an antagonist of interferon (IFN), thereby promoting viral replication [17]. The genome sequence of CoVs shows 58% identity on the nsp-coding region and 43% identity on the structural protein-coding region, suggesting that structural proteins are necessary in adaptation to new hosts [1].

## 3. Treatment and Prophylaxis against COVID-19

The binding of the virus to its host cell through its specific receptor is a crucial step for the viral infection and therefore a promising target for the development of treatments against CoVs. Evidence from previous studies suggests that the ACE2 gene, which encodes the angiotensin-converting enzyme-2, is the host receptor for SARS-CoV-2, which is similar to SARS-CoV-1. The connection occurs through the S protein and is 10–20-fold greater compared to SARS-CoV-1, which may explain the high transmission rate of SARS-CoV-2 in humans [16,17]. ACE2 is predominantly found in epithelial cells of the lower respiratory tract as well as the enterocytes of the human intestine [17]. At present, as there is not a specific therapy against CoV infections, most treatments are supportive and include managing complications a patient develops, empirical antimicrobial drugs, antipyretics/analgesics, mechanical ventilation, oxygen therapy, conservative fluid supply and corticosteroids if indicated for other reasons [18]. Here, we summarize the main experimental therapeutics available for the treatment of CoVs.

Antiviral drugs: Remdesivir is a nucleotide analogue with a broad spectrum of antiviral activity against RNA viruses; it blocks the RNA-dependent RNA polymerase which inhibits the virus replication [18,19]. However, its effectiveness and safety have not been tested in clinical trials yet. Ribavirin is another nucleoside analogue of broad-spectrum antiviral activity which inhibits RNA synthesis. Despite studies demonstrating that this drug inhibits SARS-CoV-1, MERS-CoV, and HCoV-OC43 in vitro, tested doses were significantly higher than typical concentrations used in humans [20,21]. Lopinavir/ritonavir are used in the treatment against HIV infection. Previous studies showed a good therapeutic effect in SARS and MERS, thus these drugs have been recommended for clinical treatment of COVID-19 [22,23]. Arbidol is a broad-spectrum antiviral agent commonly used in influenza virus infections. In vitro assays have shown antiviral activity against SARS-CoV-1 and SARS-CoV-2 [24].

Antiviral peptides: previous reports have described some potential peptides against SARS-CoV-1 and SARS-CoV-2 infections, these target the S protein–ACE2 interaction and entry process [25,26,27]. Its main advantages are associated with specificity, affinity and low cytotoxicity; however, one of its disadvantages is its short half-life in vivo [28].

Antimalarial agents: Chloroquine/Hydroxychloroquine: In vitro studies have shown its potent antiviral effect against the SARS-CoV-2, preventing viral replication by interfering with the ACE2 receptor. However, chloroquine is a drug with limited use due to its resistance as well as toxicity [29].

Monoclonal antibodies (mAbs): mAbs can neutralize SARS-CoV-1 and inhibit formation of epithelial syncytia through interfering with the binding of some structural proteins with its receptor [30]. In vitro and in vivo studies by multiple groups identified mAbs targeting either SARS-CoV-1, SARS-CoV-2 or MERS-CoV that inhibited viral replication [31,32]. Nevertheless, a limitation for the development of mAbs is the long period of time until their clinical application.

Human convalescent plasma: Convalescent plasma from patients who have recovered from viral infections has been used as immunotherapy to SARS and COVID-19 [33,34]. Studies have suggested that plasma contains high titres of neutralizing antibodies that may suppress viremia if administered early in the course of infection [35].

Vaccines: The best strategy for prevention of new outbreaks of CoVs is the development of a vaccine providing protective active immunity. There are a large number of proposed vaccines against SARS-CoV-2, based on the following strategies: inactivated viruses, live-attenuated viruses, viral vector-based vaccines, subunit vaccines, recombinant proteins and DNA/RNA vaccines. Some of them have been approved for emergency use by the WHO such as Pfizer/BioNtech, Astrazeneca/AZD1222, Janssen/Ad26.COV 2.S, Moderna, Sinopharm and Sinovac-CoronaVac.

## 4. Challenges in Developing CoV Vaccines

Over the past two decades, new outbreaks of infectious diseases have emerged, e.g., Zika, Ebola, MERS, SARS and now COVID-19. With each new threat there is an urgent need to develop safe and effective vaccines within a short period of time. The effectiveness of a vaccine depends on the properties of antigen recognition, activation, expansion, memory, vaccine trafficking and a multitude of specialist functions of lymphocytes [36]. An ideal vaccine should induce and maintain efficacious concentrations of virus-specific antibodies, as well as specific T-cell immunity, without causing significant adverse effects. Nevertheless, historically vaccine development has always been a slow process, generally taking years to produce a licensed vaccine [37]. Therefore, an ideal production platform should have the following characteristics: speed, low cost and high scalability, while it should demonstrate elicitation of consistent immune responses across pathogens [38]. The use of new technologies such as next-generation sequencing and reverse genetics can help shorten the time of vaccine development [38]. Even so, there are several challenges to overcome for the development of an efficient vaccine against CoVs, regardless of production system and final formulation.

At first, the optimization of the design of the antigen that will act as a potent immunogen is a crucial step to ensure the most efficacious immune response. Based on studies of Respiratory Syncytial Virus (RSV) vaccine development, it seems that the most protective epitopes of the S protein together with a Th1 T cell response may be key features of a safe SARS-CoV-2 vaccine [39]. Indeed, the S protein of SARS-CoV-2 has been shown to be a promising immunogen to induce protection, so almost all SARS-CoV-2 vaccine candidates, including plant-produced ones currently under development are targeted to this surface protein. Some reports [40] have identified a dimeric form of the receptor-binding domain (RBD) of the S protein of MERS-CoV as a potent immunogen due to increased antibody titers compared to the conventional monomeric form. This knowledge could be applied for the development of CoVs vaccines not only against MERS but SARS-CoV-1 and 2 too. However, the debate continues whether the best approach is to include the full-length S protein or only RBD [41,42]. Perhaps incorporating more than one antigen into the vaccine formulation could have a positive effect by extending its immunogenicity. Safety is another primary consideration of any vaccine and previous studies with vaccine candidates for SARS and MERS having raised concerns over exacerbating the humoral response associated with adverse effects [38,39]. Hence, testing in a suitable animal model permissive to viral replication and that develops pathologic and clinical features consistent with the human disease is a strictly rigorous step. Another concern is that the duration of immunity has not been determined, so it is uncertain if single-dose vaccines will confer immunity. Some developers of licensed SARS-CoV-2 vaccines such as Moderna and Pfizer have considered adding a third dose to its immunization scheme in order to increase the titers of neutralizing antibodies, thereby ensuring the protective efficacy against SARS-CoV-2 variants. The vaccine formulation including the immunogen plus adjuvant can influence the type of immune response. In the case of vaccines based on subunits or inactivated antigens, the addition of adjuvants for directing the types and magnitude of immune responses is necessary. In pre-clinical CoV vaccine studies, adjuvants such as aluminum salts, emulsions, and toll-like receptor (TLR) agonists have been used in the vaccine formulations [43]. Among these adjuvants, aluminum salts have been applied in S protein or RBD vaccine formulations which have shown to induce neutralizing antibody production [44,45]; these have been associated with protection against SARS-CoV-2 infection. However, alum lacks the capability to promote the activation of CD4+ and CD8+ T cell responses, which together with the antibody responses provide a protective immunity against the SARS-CoV-2 [46]. Other adjuvants, e.g., emulsion adjuvants and TLR agonists, could play a more favorable role inducing both humoral and cellular immune responses [43]. The formulation can also be associated with the administration route. So far, all implemented or tested vaccines against SARS-CoV-2, regardless of their nature (Table 1), are delivered via intramuscular or intradermal injection. Yet intranasal or inhaled application is considered through mucosal membranes of the respiratory tract to activate the mucosa-associated lymphoid tissue (MALT), especially its parts, NALT (Nasal ALT) or BALT (Bronchus ALT), and induce a strong local immune response at the site of the virus invasion [47]. The above described challenges concern all vaccines against SARS and MERS, thus also subunit or VLP-based vaccines (see below) possible to produce in plant systems, but analogously to other vaccines—purified and then administered. In the case of oral vaccines, often identified with plant-derived ones, the challenge itself remains the barrier of GALT (Gut ALT) functioning associated with oral tolerance mechanisms (see Section 5).

Regardless of particular approaches to vaccine development, with the COVID-19 pandemic expanding globally and new mutants of SARS-CoV-2 emerging the progress in vaccine production and availability is highly required. Currently there are at least 68 SARS-CoV-2 vaccines under clinical trials. Studies of other related coronavirus such as SARS and MERS can provide important information on aspects of immunity and protection as well as identify features that merit attention with respect to vaccine safety. Protection against SARS-CoV-2 challenge have been demonstrated in preclinical studies in non-human primates [48,49,50,51,52]. However, more studies are necessary in order to establish both the durability of protection and the vaccine safety. Current efforts in vaccine development against SARS-CoV-2 could provide key answers concerning immune control of COVID-19 infection and possibly against new outbreaks of pathogenic coronaviruses in the future.

### 4.1. Approaches for Anti-CoV Vaccines

Classical vaccines are based on live-attenuated pathogens [53,54,55,56], inactivated pathogens [39,55] or only specific fragments of pathogens (subunit vaccines) [36,57,58]. However, in recent years new vaccine platforms have been developed with more rapid timelines and with a more easily modifiable antigen design. These systems include virus-like particles (VLPs) [59,60,61,62] and vaccines based on DNA [59,60,61] or RNA, including modRNA—containing naturally modified nucleosides or synthetic nucleoside analogs [60,61,62,63].

In Table 1 we summarize advantages/disadvantages of each strategy and present some COVID-19 vaccines made with these technologies.

**Table 1 vaccines-10-00138-t001:** Overview of Vaccine Production Platforms for COVID-19.

Vaccine Platform	Advantages	Disadvantages	Developers (Vaccines)	Reference
Live attenuated	Mimic natural infection Long-lasting immunity Process used for several licensed human vaccines	Possibility to reverse to natural form Contraindicated in immunocompromised individuals	Codagenix (COVI-VAC) Meissa Vaccines, Inc. (MV-014-212)	[53,54,55,56,57,58,59,60,61]
Inactivated	No replication of the inactivated pathogen High stability Process used for several licensed human vaccines	Antigen and/or epitope integrity needs to be confirmed Multiple booster doses are required to obtain long-term protection	Sinovac (CoronaVac) Sinopharm (BIBP-CorV) Osaka University (CovidVax)	[39,55]
Subunit	Non-infectious Safe	Reduced immunogenicity, adjuvants are often needed	Novavax (NVX-CoV2373) Sanofi Pasteur (VAT00008) Instituto Finlay de Vacunas (Soberana 02) Kentucky Bioprocessing (KBP-201)	[36,57,58]
VLPs	Multimeric presentation of antigen Safe No viral replication	Purification can be a limiting factor	Astrazeneca (Vaxzevria) CanSino Biologics Inc. (Convidicea) Gamaleya Research Institute (Sputnik V) Janssen Pharmaceuticals (Ad26.COV2-S)	[59,60,61,62]
DNA	Induce both humoral and cell-mediated immune responses Highly scalable	Low immunogenicity	Zydus Cadila (ZyCoV-D) Inovio Pharmaceuticals (INO-4800)	[59,60,61]
RNA and modRNA	Safe Low-cost and highly scalable	Limited experimental information Instability	Pfizer/BioNTech (BNT162b2) Moderna (mRNA-1273) CureVac AG (CVnCoV) Imperial College London	[61,62,63]

### 4.2. Molecular Farming for Recombinant Protein Expression

Over the last decades, plant-based expression systems have emerged as a novel platform for the production of recombinant proteins in view of a number of advantages compared to the traditional systems (bacteria, yeast species, insect or mammalian cells and transgenic animals). These advantages include rapid production, high scalability, low cost, safety and the capacity to produce multimeric or glycosylated proteins [64,65]. Until now, over 100 recombinant proteins such as human serum proteins, growth regulators, vaccines, cytokines, antibodies and enzymes have been produced in different plant species and several of them have reached the late stages of commercial development [66,67,68]. Importantly, in 2012 “Elelyso”, a recombinant enzyme produced in carrot cells and commercialized by Protalix Biotherapeutics (Karmiel, Israel), was approved by the Food and Drug Administration (FDA) for treatment of Gaucher’s disease [69]. Other two products which have been licensed are: (1) the plant made scFV mAb used in the production of a recombinant HBV vaccine in Cuba, and (2) the Newcastle disease virus (NDV) vaccine for poultry approved by the US Department of Agriculture (USDA) [70]. However, there are several vaccines produced by plants currently at the clinical trial phases, e.g., VLP-based vaccines against influenza produced by Medicago Inc. (Quebec City, QC, Canada) [71].

Due to its easy genetic transformation and rapid development, *Nicotiana benthamiana* and *N. tabacum* are the two species most commonly used for the expression of recombinant proteins in plants. Nevertheless, there are many cereal crops, fruits, and vegetables such as rice, maize, soybean, lettuce, tomato, carrot, potato, and alfalfa that have also been evaluated in plant molecular farming [72]. Interestingly, the use of edible plants for vaccine production had led to the introduction of the term “edible vaccines”. This concept evolved through the years into “oral vaccines” obtained via at least some plant material processing and delivered under a controlled regime similar to medicines. Although efficacy of oral vaccines still needs to be improved, these could be considered useful, especially in developing countries thanks to the lower production costs and no specialist equipment or facilities required for their storage and application [73].

#### Plant Transformation Approaches

The expression methods used in molecular farming are of the stable or transient type. Both methods are being used in the production of antigens for the development of vaccines against COVID-19. The first can be further subdivided into techniques involving *Agrobacterium*-mediated transformation of the nuclear genome or the biolistic (particle bombardment) method enabling stable modification of nuclear or chloroplast (plastid) genomes, whereas transient expression may be achieved using viral vectors or by infiltration with *Agrobacterium* carrying dedicated vectors [74]. These expression approaches are summarized in Figure 2.

Stable methods comprise the transgene insertion into a nuclear or chloroplast genome. A majority of the recombinant proteins to date have been produced by nuclear *Agrobacterium*-mediated transformation, which takes advantage of the bacteria’s ability to transfer DNA segments into the host genome, thereby conferring heritable traits to the progeny. Alternatively, although much less frequently, plants are transformed using the biolistic method. However, in both cases the insertion of the transgene via heterologous recombination is random, which can lead to positional effects such as disruption of essential genes, silencing and unpredictable levels of expression in obtained transgenic plants [75]. Another concern is the potential risk of genetically modified plants crossing with native species or food crops. These methods have also been used to express recombinant proteins in dry seeds of cereals, which prolongs their half-life since no cold chain is required for their conservation [74]. Additionally, systems based on transgenic plants have a high scale-up capacity and they have been used for the production of many oral vaccines, which may have lower production costs thanks to the fact that the antigen purification process is not necessary [75].

Alternatively, plastid transformation focuses on expressing the transgenes in chloroplasts. This technology involves the use of a biolistic process in which microparticles of gold or tungsten are coated with foreign DNA, placed on a gene gun and fired at high pressure. Next, the coated DNA travels at a high speed within a vacuum and penetrates into the cells of the targeted plant [76]. Although the biolistic method can be used both for chloroplast and nuclear transformation, the main difference is based on the site-directed insertion of a transgene together with the defined flanking sequences into the plastid genome (plastome) via homologous recombination instead of random integration in the case of the nuclear genome. Besides, transplastomic plants represent additional advantages over transgenic ones, which includes a high recombinant protein yield because of the high number of DNA copies per plastid and then plastids per a vegetal cell, expression of polycistronic genes, natural transgene containment since plastid genes are not usually transmitted through pollen and lack of position effect/gene silencing [75]. The main disadvantage of this system is connected with the fact that it is yet to be adapted for many plant species apart from tobacco and lettuce.

Transient expression systems possess multiple advantages compared with stable methods. These offer rapid and high-protein expression within a few days, hence are considered as a suitable convenient platform, especially for the production of vaccine antigens or antibodies during a pandemic situation. One approach of this system is based on the use of plant viruses as vectors to deliver foreign genes without integration into the plant genome. The main viruses used here are these of the RNA type, such as tobacco mosaic virus (TMV), potato virus X (PVX), alfalfa mosaic virus (AlMV), cucumber mosaic virus (CMV) and cowpea mosaic virus (CPMV) [77]. Several publications have reported the successful expression of different immunogenic epitopes as fusions to viral coat proteins. Generally, the limit imposed by ‘full virus’ vectors on the size of the insert is about 1 kb [78]. Another, at present more common approach is agroinfiltration—delivery of mixed vectors composed of elements coming from plant viruses and *Agrobacterium* vectors. This process is in essence an infiltration of whole mature plants with a diluted suspension of *Agrobacterium* carrying T-DNAs encoding RNA replicons [78]. The infiltration of plants is usually achieved by vacuum infiltration for 10–30 s or by syringe infiltration [79]. This strategy combines the advantages of three biological systems: the speed and expression level/yield of the virus, the transfection efficiency of *Agrobacterium*, and the posttranslational capabilities and low production cost of plants [78]. Depending on the vector used, the host organism and the initial density of bacteria, the process takes from four to ten days and the expression levels depending on the nature of the gene of interest can reach a yield of up to 5 g recombinant protein per kg of fresh leaf biomass or over 50% of total soluble protein. Additionally, this system has the capacity to express longer genes, up to 2.3 kb inserts or up to 80 kDa proteins [78].

### 4.3. Plant-Made Vaccines against COVID-19 and SARS

Molecular farming constitutes a consolidated platform for the manufacturing of biopharmaceuticals. At present, this technology is applied for the formulation of injectable or nasal vaccines against different infectious diseases, some of which have entered advanced stages of clinical trials [80]. There are also some examples of plant vaccines directed against COVID-19 and SARS, as summarized in Table 2.

Li et al. (2006) successfully expressed a segment (sequence encoding amino acids 1-658) of the SARS-CoV-1 S protein in nuclear- and chloroplast-transformed plants. Western blot analysis revealed the expression and accumulation of recombinant S protein in transient and stable transgenic or transplastomic plants [81].

Zheng et al. (2009) produced the SARS-CoV-1 nucleocapsid protein in *N. benthamiana* plants via transient expression, using the post-transcriptional gene silencing suppressor p19 protein from tomato bushy stunt virus to enhance the expression of recombinant protein. The yield was 79 µg per g fresh leaf weight at three days post infiltration. Moreover, BALB/c assays of mice intraperitoneally vaccinated with a pre-treated plant extract emulsified in Freund’s adjuvant induced high levels of IgG1 and IgG2a antibodies and upregulation of IFN-γ and IL-10 in splenocytes [82].

Pogrebnyak et al. (2005), expressed the N-terminal fragment of SARS-CoV-1 S protein (S1) in tomato and low-nicotine tobacco plants. The plant-derived antigen was evaluated in vivo, and mice showed significantly increased levels of SARS-CoV-specific IgA after oral ingestion of tomato fruits expressing the S1 protein. Sera of mice parenterally primed with tobacco-derived S1 protein revealed the presence of SARS-CoV-specific IgG as detected by Western blot and ELISA analysis [83].

Ward et al. (2021) reported the first clinical study of a VLP vaccine (CoVLP) using the full-length S glycoprotein produced by transient expression in *N. benthamiana* plants. The vaccine formulations consisted of CoVLP unadjuvanted or adjuvanted with either CpG 1018 (composed of 227 cytosine-phosphoguanine (CpG) motifs) or AS03 (an oil-in-water emulsion containing tocopherol and squalene). The vaccine formulations were administered in two doses, 21 days apart, to healthy adults 18–55 years of age. Results showed that unadjuvanted CoVLP had the lowest reactogenicity and was recognized by the immune system after the second dose, but the immune responses were modest [84]. In the case of adjuvants incorporated in this study, enhancement of humoral and cellular responses to the S protein and in promoting responses at lower CoVLP doses, AS03 appeared to be more effective than CpG1018.

Kentucky BioProcessing Inc. (KBP) has also been developing a COVID-19 Subunit Vaccine (KBP-201) using the agroinfiltration technique in *N. benthamiana* plants. KBP reported that its candidate vaccine showed a positive result with stimulation of immune responses in its pre-clinical trials [85]. Currently KBP-201 is under phase 1/2 of clinical trials, the results of which are yet to be published. iBio, a Texas based biotherapeutics company, is developing two potential COVID vaccines, glycosylated IBIO-200 and non-glycosylated IBO-201. The latter is a carrier protein-conjugated subunit vaccine generated by the fusion of the S protein with LicKM, which is an engineered thermostable lichenase enzyme, originally from *Clostridium thermocellum*. Conjugation of a target antigen with LicKM increases the expression, stability and immunogenicity of the vaccine [86]. In pre-clinical trials both vaccines were reported to show the ability to stimulate the immune system and they produced anti-spike specific antibodies, with IBIO-201 showing more promising results, with higher levels of neutralizing antibodies against the SARS-CoV-2 virus [85].

Baiya Phytopharm, Thailand, is using the BaiyaPharming™ technology to express the spike protein in *N. benthamiana* plants as a COVID-19 vaccine (Baiya SARS-CoV-2 Vax 1). Preclinical results demonstrated the production of neutralizing antibodies in mice and monkeys after two doses of purified protein [87]. Based on these positive results, the company is currently assessing the toxicology and side-effects of the vaccine.

## 5. Future Directions for Anti-CoV Plant-Made Vaccines

The fast spread of SARS-CoV-2 suggests that vaccines will be indispensable to end or at least control this global pandemic. Fortunately, now there are many available technologies showing the potential to fight this virus. In this sense, plant biotechnology offers potential solutions through the development of highly valuable recombinant proteins such as vaccines, antibodies and antiviral peptides. Plants have been used as a platform for the production of biopharmaceuticals for more than 30 years, an approach often described as ‘molecular farming’, which has several advantages over other expression systems, such as economy, scalability and safety. Plant systems can also produce proteins of favorable glycan configurations—either slightly different but tolerable by the human immune system, or easily convertible to human type glycans, or finally—humanized [88].

Of greatest relevance, when transient expression systems are used, is that they can be scaled up rapidly to meet sudden and unforeseen demand, which makes it an ideal platform for the production of vaccines in the case of pandemics such as the one we are currently experiencing. The success of this approach is evidenced by several molecular farming companies that specialize in the development of plant-derived proteins, for example Medicago Inc. (Quebec City, QC, Canada), Agrenvec (Madrid, Spain), Protalix BioTherapeutics (Karmiel, Israel), Diamante (Verona, Italy), ORF Genetics (Kópavogur, Iceland), Fraunhofer (Newark, DE, USA) and Ventria Bioscience/Invitria (Fort Collins, CO, USA). Two strategies that are being developed as a means to address the COVID-19 pandemic are the production of subunit vaccines based on individual proteins or VLPs with multiple copies of SARS-CoV-2 antigens arrayed on their surface. Both strategies have been well-studied in plants and several subunit vaccine candidates have already been obtained [89]. In the case of SARS, a prototype vaccine was manufactured within three weeks after receiving the protein sequences, with up to 200 mg of protein produced per kg of fresh leaves [83]. At least one company is thought to be developing a COVID-19 vaccine based on the expression of SARS-CoV-2 protein subunits in tobacco plants, namely Kentucky BioProcessing (Owensboro, KY, USA) [85]. As an alternative to subunit vaccines, the development of VLPs has multiple advantages, since the ordered antigen arrangement can trigger even stronger cellular and humoral responses. Medicago Inc. (Quebec City, QC, Canada) has announced the production of VLPs against SARS-CoV-2 by a transient expression system in tobacco plants. This company has estimated the production capacity of 10 million doses per month of VLPs-based vaccines.

VLPs composed of SARS-CoV-2 proteins, mostly S, and produced using the transient expression technology can be considered as the first choice in the use of plant biotechnology to combat COVID-19 pandemics. However, alternatively to VLPs composed by whole or slightly truncated SARS-CoV-2 proteins, only the key virus epitopes, when identified, can potentially be used as vaccine immunogens. In this approach, these epitopes could be covalently or non-covalently attached to carrier VLPs, especially these assembled by HBcAg (Hepatitis B core Antigen) [90,91]. Prominent HBcAg properties of various epitope presentations result in the stability of chimeric ‘monoepitopic’ VLPs, composed of solely HBc-epitope monomers. Durability of VLPs displaying heterologous epitopes may be further enhanced via the mosaic structure of VLPs. This VLP form makes it possible to reduce steric hindrance in the case of epitopes of particularly large size or specific structure. So far mosaic HBcAg VLPs have been obtained in plant expression systems via the assembly of HBc+HBc-epitope heterodimers [92]. However, an alternative method may be proposed based on the co-expression of an unmodified HBcAg and HBc-epitope, analogously to heteromultimeric VLPs of the bluetongue virus [93]. Chimeric HBcAg VLPs of both types were produced via transient expression, but also in transgenic or transplastomic plants. Regardless of the expression system and VLP type, displayed epitopes of viral, bacterial and even parasite origin showed proper immunogenicity, e.g., the HBE capsid epitope, the ZIKV E protein domain III (zDIII), M2e of influenza virus, domain-4 of the anthrax protective antigen (PA-D4) or cysteine protease of liver fluke [94,95,96,97,98]. Hence, considering the current state-of-the-art technology, it can be stated that there is no technological barrier to obtaining plant-produced monoepitopic or mosaic VLPs with a cluster of epitopes as vaccines against a given SARS-CoV-2 variant or even a multivalent vaccine against several variants of the virus or various coronaviruses.

Apart from HBcAg, other VLPs coming from human or animal viruses can also be used as epitope carriers, but to date only some have been produced in plants, e.g., these assembled by S-HBsAg (Small Hepatitis B surface Antigen) or HPV (Human Papilloma Virus) L1 protein [99,100,101]. However, in recent years VLPs derived from plant viruses such as e.g., TMV, CMV or PapMV, (Papaya Mosaic Virus) or PVX, are more and more extensively investigated both as epitope carriers and for other biopharming and nanotechnology purposes [102,103]. Moreover, such VLPs can be safely produced in substantial quantities, due to capacity of massive propagation of plant viruses in suitable hosts [70]. Considerable potential as an epitope carrier is also reported to exist for the oligomeric proteins LTB (heat-labile enterotoxin) and CTB (cholera toxin B), which at the same time act as mucosal adjuvants, as confirmed also in the case of plant-expressed LTB and CTB [104]. The use of carrier platforms may facilitate rapid development of new vaccine variants in response to emerging virus mutants. All these VLPs or oligomers can be used for injection after purification, but prospectively also as vaccines delivered through mucosa, i.e., as intranasal, inhaled or sublingual vaccines (Figure 3).

Furthermore, since the function of particular parts of MALT such as e.g., NALT or BALT share analogous functioning mechanisms to GALT and they interact with each other, it can be assumed that oral vaccination against SARS-CoV-2 and other coronaviruses would also be developed over time [105,106]. Production of oral vaccines is based on stable antigen expression in transgenic or transplastomic plants and requires only partial tissue processing, usually involving lyophilization [107]. However, development of an appropriate oral administration regime is particularly important. Antigen dosage, frequency of delivery and adjuvants have to be meticulously adjusted to induce efficacious mucosal and systemic responses instead of oral tolerance acquisition [106,108]. Nevertheless, plant-derived oral vaccines were demonstrated to be fully efficacious when applied as boosting doses [109]. Therefore, even in that form they may still be useful at the post-pandemic stage, especially if the SARS-CoV-2 virus becomes a seasonal pathogen. A low-cost plant-derived vaccine would be an advantageous alternative, particularly in developing countries (Figure 3).

## 6. Conclusions

Pandemics will generate simultaneous demand for vaccines around the world. Therefore, the use of different platforms for the production of safe and effective vaccines is desirable in order to meet global demand. In this sense, the advantages of plant-based platforms such as low cost, speed, scalability, and safety could help to cover the global demand. However, knowledge of the use of plant expression systems for vaccine production should be more widely disclosed to promote their adoption by governments or private companies in prospect to increase the global health; especially in developing and low-income countries. Discussions with global stakeholders about organizing and financing large-scale vaccine manufacturing, procurement, and delivery are required.

## Figures and Tables

**Figure 1 vaccines-10-00138-f001:**
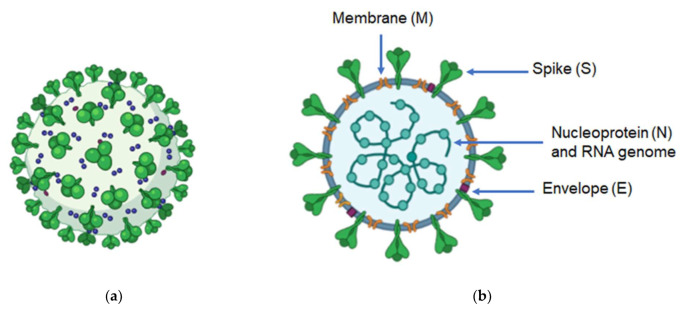
General structure of coronavirus. (**a**) 3D image; green club-shape figures: spike protein, blue dots: membrane protein, red dots: envelope protein (**b**) main structural proteins of CoVs. Created with BioRender.com.

**Figure 2 vaccines-10-00138-f002:**
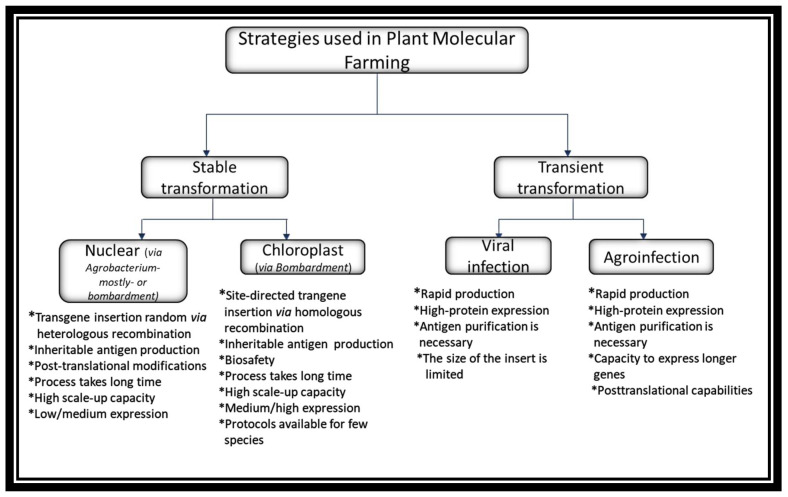
Plant expression systems for biopharmaceutical production.

**Figure 3 vaccines-10-00138-f003:**
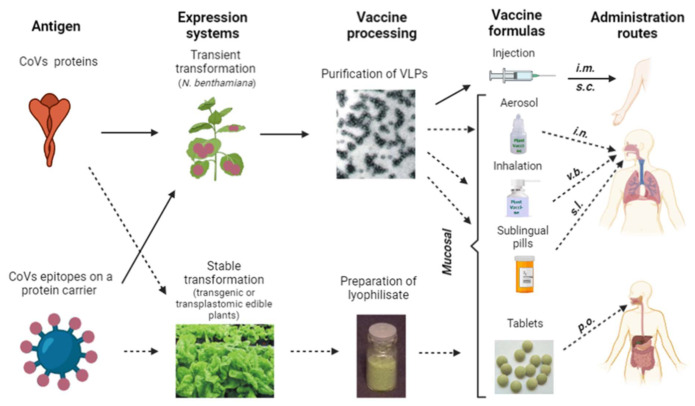
Summary of currently exploited (full arrows) and potential future (dotted arrows) approaches to manufacturing and application of various types of plant-based vaccines against CoVs. Administration routes: *i.m.*—intramuscular, *s.c.*—subcutaneous, *i.n.*—intranasal, *v.b.*—via bronchi, *s.l.*—sublingual, *p.o.*—per os.

**Table 2 vaccines-10-00138-t002:** Approaches used for the production of plant vaccines directed against CoVs.

Approach	Expressed Antigen/ Administration Route	Relevant Results	Reference
Nuclear and chloroplast transformation	1-658 amino acids of SARS-CoV-1 S protein	The antigen was successfully expressed in transgenic tobacco and lettuce as well as in transplastomic tobacco.	[80]
Transient expression	Recombinant SARS-CoV-1 nucleocapsid (rN) protein/intraperitoneal	p19 protein enhanced the transient expression of rN up to a concentration of 79 µg per g fresh leaf weight, which induced in mice high levels of IgG1 and IgG2a.	[81]
Nuclear expression	N-terminal fragment of SARS-CoV S protein (S1)/oral	S1 protein was expressed in tomato and low-nicotine tobacco plants, which induced specific IgA and IgG responses in mice.	[82]
Transient expression	Full-length S glycoprotein of SARS-CoV-2/intramuscular	CoVLP alone or adjuvanted with either CpG1018 or AS03 suggests that the candidate vaccine is well-tolerated and immunogenic. Its immunogenicity, particularly at low doses, is radically enhanced by the presence of an adjuvant.	[83]
Transient expression	Protein subunit vaccine based on the SARS-CoV-2 receptor-binding domain (RBD)/intramuscular	The vaccine showed a positive result on stimulation of immune responses in pre-clinical trials. Clinical trials results not yet published.	[84]
Transient expression	Subunit vaccine combining antigens derived from the SARS-CoV-2 spike protein fused to LicKM/intramuscular	In pre-clinical trials the vaccine (IBIO-201) stimulated the immune response producing high titers of neutralizing antibodies.	[85]
Transient expression	Subunit vaccine from SARS-CoV-2 spike protein/injection (no more data provided)	The vaccine was able to induce antigen-specific IgG and neutralizing responses as well as cellular immunity in animals.	[86]

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
