# Peer review of "Plant-Based Vaccines in Combat against Coronavirus Diseases"

_vaccines, 2022, doi:10.3390/vaccines10020138_

Round 1

Reviewer 1 Report

This article aims to describe the applications of plants for developing effective vaccines against coronavirus diseases. One major concern is the repetition of some information throughout the manuscript. Moreover, there is huge scope to make the article more precise. Below I offer several comments regarding the manuscript. 

  1. I found this section ‘4.1. Approaches for anti-CoV vaccines’ unnecessary. I would suggest deleting this section and presenting it by a table only.
  2. This section ‘4.2.1. Plant transformation approaches' discuss the general methodology of plant-based molecular firming. This section can be focused on the techniques used to produce plant-based vaccines to combat COVID-19.
  3. The review should elaborate a little more on the challenges of developing plant-based vaccines.
  4. It might be more appropriate to make a separate section for plant-based antibodies against coronavirus diseases. 
  5. I would suggest making the ‘5. Future directions for anti-CoV plant-made vaccines’ section more focused and precise. 

Author Response

RESPONSE TO REVIEW 1

               The Authors would like to thank Reviewer for positive evaluation as well as very constructive comments and remarks, which helped to improve the manuscript. Regarding particular points, these were adjusted as listed below. Apart from required or suggested corrections and complements, some related minor self-amendments have been added. All significant changes (insertions or shifts) in the text are marked in yellow.

  1. I found this section ‘1. Approaches for anti-CoV vaccines’ unnecessary. I would suggest deleting this section and presenting it by a table only.

Response: Done. We have removed this section and show it only as a table.

  1. This section ‘2.1. Plant transformation approaches' discuss the general methodology of plant-based molecular farming. This section can be focused on the techniques used to produce plant-based vaccines to combat COVID-19.

       Response: The methods described in this section are being used for the development of    vaccines against COVID-19. We have added a short paragraph to emphasize this. In the next section 4.3. Plant -made vaccines against COVID-19 and SARS, we describe in detail each work carried out for the development of a vaccine against COVID-19 and SARS as well as the technique used and the main results.

  1. The review should elaborate a little more on the challenges of developing plant-based vaccines.

       Response: Done. Most challenges of developing plant-based vaccines against coronaviruses is common to other vaccines, hence these are commonly described in the section 4. Issues characteristic to plant-based vaccines are outlined there and described with more details in the following sections.

  1. It might be more appropriate to make a separate section for plant-based antibodies against coronavirus diseases

       Response: We think that this section is unnecessary since the review is only focused  on plant-vaccine development. The issue of plantibodies against coronaviruses deserves another paper.

  1. I would suggest making the ‘ Future directions for anti-CoV plant-made vaccines’ section more focused and precise. 

       Response: In our opinion, such a section is usually of general - conceptual character. Nonetheless, here possible approaches for anti-CoV plant-made vaccines are clearly and gradually presented. Though, thanks to Revewer’s suggestion, we added some details for better apperception of this general concept. At first, vaccines produced in transient expression systems as currently predominant approach are described: a) subunit and b) VLPs-based vaccines. Next, potential vaccines based on carriers displaying epitopes are presented:  c) VLPs of different types and coming from human/animal viruses but produced in plant systems; d) VLPs derived from plant viruses;  e) oligomeric–adjuvant proteins. All these potential vaccines are presented as used by injection and in prospect administered through mucosas. Finally, since at this moment this approach is conceptual only – VLPs produced in transgenic or tranplastomic plants as oral vaccines are outlined.

We hope that the introduced corrections and explanations will truly meet your expectations and you will find the manuscript in its revised form suitable for publication. We will sincerely appreciate the acceptance of our revised paper.

With kind regards,

Authors

Reviewer 2 Report

The manuscript “Plant-based vaccines in combat against coronavirus diseases” by Benita Ortega Berlanga and Tomasz Pniewski, is a review about the production of vaccines or components of vaccines using plants. The review suggests that plant-produced Coronavirus proteins or peptides can be applied as mucosal vaccines, or as oral vaccines obtained from partly processed plant tissue. The manuscript is well written and presented. I don´t have major concerns about the manuscript.

Minor concerns:

1) In the legend of Fig 1.a, please add the description of the elements showing in the image such as the blue dots.

2) Please add references to table 1.

3) Lines 232-233, “no structural genes” is referred twice. Please remove one.

4) In the RNA vaccines section, please include a sentence referring at the application of modified nucleosides, such as pseudouridine and others.

5) The last paragraph should be an overall conclusion from the review, and not finishing with “figure 3”. Please include a final paragraph, with a general conclusion and/or suggestions for the future.

6) In the lines 461-462, please remove the sentence “Yet in total it takes a longer time, mostly due to necessary adjustment of the antigen-specific formulation and dosage”. This information is vague and can be applied to any vaccine under development.

Author Response

RESPONSE TO REVIEW 2

               The Authors would like to thank Reviewer for positive evaluation as well as very constructive comments and remarks, which helped to improve the manuscript. Regarding particular points, these were adjusted as listed below. Apart from required or suggested corrections and complements, some related minor self-amendments have been added. All significant changes (insertions or shifts) in the text are marked in yellow.

  1. In the legend of Fig 1.a, please add the description of the elements showing in the image such as the blue dots.

Response: Done.

  1. Please add references to table 1.

Response: Done.

  1. Lines 232-233, “no structural genes” is referred twice. Please remove one.

Response: Done.

  1. In the RNA vaccines section, please include a sentence referring at the application of modified nucleosides, such as pseudouridine and others.

Response: At the suggestion of reviewer 1 we have removed this section and present it only as a table. However the modRNA approach is mentioned in the introduction to the table and in the table too.

  1. The last paragraph should be an overall conclusion from the review, and not finishing with “figure 3”. Please include a final paragraph, with a general conclusion and/or suggestions for the future.

Response: Done. A general conclusion has been included.

  1. In the lines 461-462, please remove the sentence “Yet in total it takes a longer time, mostly due to necessary adjustment of the antigen-specific formulation and dosage”. This information is vague and can be applied to any vaccine under development.

Response: Done. The sentence has been deleted. However, as it had been used in the aspect of plant-derived oral vaccines, we slightly changed the following sentences to maintain the context.

We hope that the introduced corrections and explanations will truly meet your expectations and you will find the manuscript in its revised form suitable for publication. We will sincerely appreciate the acceptance of our revised paper.

With kind regards,

Authors

Round 2

Reviewer 1 Report

The paper ‘Plant-based vaccines in combat against coronavirus diseases ‘  has improved and may be accepted for publication in the present form.